# Application of FTIR Spectroscopy for Quantitative Analysis of Blood Serum: A Preliminary Study

**DOI:** 10.3390/diagnostics11122391

**Published:** 2021-12-18

**Authors:** Lyudmila V. Bel’skaya, Elena A. Sarf, Denis V. Solomatin

**Affiliations:** 1Biochemistry Research Laboratory, Omsk State Pedagogical University, 644099 Omsk, Russia; belskaya@omgpu.ru; 2Department of Mathematics and Mathematics Teaching Methods, Omsk State Pedagogical University, 644043 Omsk, Russia; denis_2001j@bk.ru

**Keywords:** FTIR spectroscopy, blood serum, modeling, biochemistry

## Abstract

The aim of this study was to analyze the possibility of simultaneous determination of the concentration of components from the characteristics of FTIR spectra using the example of a model blood serum. To prepare model solutions, a set of freeze-dried control sera based on bovine blood serum was used, certified for approximately 38 parameters. Based on the values of the absorbance and areas of absorption bands in the FTIR spectra of model solutions, a regression equation was constructed by solving a nonlinear problem using the generalized reduced gradient method. By using the absorbance of the absorption bands at 1717 and 3903 cm^−1^ and the areas of the absorption bands at 616, 3750, and 3903 cm^−1^, it is possible to simultaneously determine the concentrations of 38 components with an error of less than 0.1%. The results obtained confirm the potential clinical use of FTIR spectroscopy as a reagent-free express method for the analysis of blood serum. However, its practical implementation requires additional research, in particular, analysis of real blood serum samples and validation of the method.

## 1. Introduction

Plasma and serum remain the main clinical specimens of interest. They contain over 300 types of proteins, as well as carbohydrates, lipids and amino acids and over 100,000 metabolites in various concentrations [1]. In addition to a rich source of biomarkers for the diagnosis of diseases, an imbalance of endogenous components of plasma and serum is of great clinical importance [2]. Determination of the clinical parameters of serum and blood plasma is widely used for the diagnosis of various diseases, as well as a way to monitor the effectiveness of treatment [3]. As a result, the demand for clinical assays is growing, leading to the use of automatic analyzers, many of which are based on colorimetric reactions and ELISA determinations. These methods involve the use of expensive and specific reagents. Given the widespread nature of such studies, there is a clear need for new and cheap alternative analytical procedures, especially as screening tools in situations where economic resources are limited and diagnostic evidence is required at the point of care.

For these purposes, vibration spectroscopy methods, in particular infrared (IR) spectroscopy, can be used, since they do not require marking, are economical, easy to operate, and require minimal sample preparation. Their sensitivity to subtle changes in biochemical composition makes them ideal diagnostic tools, and recent advances in technology and data analysis enable fast and non-invasive analysis of body fluids [4,5,6,7,8,9]. However, the determination of clinical parameters in sera using vibration spectroscopy can be difficult due to the high complexity of the matrix and the low concentration of some analytes, since chemometric algorithms are usually required to eliminate matrix effects [10,11]. In general, the idea of determining the clinical parameters of serum from IR spectra is not new and the potential for quantitative and semi-quantitative analysis of metabolites in plasma, serum and whole blood has been confirmed by a number of studies [12,13,14,15]. However, modeling of blood serum has not been performed to date.

In this work, we modeled normal blood serum (SPINREACT normal, 38 components) and analyzed the possibility of simultaneous determination of the concentration of components from the characteristics of IR spectra.

## 2. Materials and Methods

### 2.1. Preparation of Model Solutions

To prepare model solutions, we used a set of freeze-dried control sera based on bovine blood serum, certified for approximately 38 parameters (SPINREACT, S.A. Ctra. Santa Coloma, Girona, Spain). The original model serum solution was diluted with bidistilled water 2, 3, 5, 7, and 10 times (in triplicate).

### 2.2. Receiving and Processing of IR Spectra

Samples of model solutions with a volume of 50 μL were applied to a zinc selenide substrate and dried in an oven at 37 °C for 60 min. The infrared absorption spectra were registered in the range of 500–4000 cm^−1^ using an FT-801 Fourier IR spectrometer (Simex, Saint Petersburg, Russia). Spectra were recorded with a scan number of 32 and a resolution of 4 cm^−1^. A background (air) measurement was taken for every sample processed. The peaks corresponding to CO_2_ vibrations were removed using the “straight line generation” option in the ZaIR 3.5 software (Simex, Russia). Three spectra were compared for each sample. The results were presented as an averaged spectrum. ZaIR 3.5 software (Simex, Russia) was used to carry out baseline correction and normalization of FTIR spectra. Raw spectra were pre-processed using a simple two-point linear subtraction baseline correction method. Spectra were then vector normalized.

We selected absorption bands that are present in the FTIR spectra of serum regardless of dilution. Absorbance (H) and area (S) were determined for the respective absorption bands.

### 2.3. Regression Model Development

At the first stage, S and H were selected in such absorption bands, which showed a high correlation with the concentrations of the components (*B_i_*). They turned out to be H20, H49, S2, S42, S49, where H20 and H49 are the absorbance of the absorption bands at 1717 and 3903 cm^−1^; S2, S42 and S49 are the areas of absorption bands at 616, 3750 and 3903 cm^−1^. At the second stage, a program was written in Delphi and such a combination of *K* of these parameters was selected, which has a high correlation with *B_i_* with a change in concentration (1, 2). Since the operations of multiplication and division of variables can enhance correlations, and multiplication by a constant, addition and subtraction of variables does not affect the correlations. As a result, from all combinations of *K*, those for which the correlation with each *B_i_* is strong (Pearson’s correlation coefficient r > 0.999) had their weighted average sum compiled, which has the minimum sum of squares of the deviation of values from *B_i_*. The vector of weight coefficients in this sum was sought by solving a nonlinear problem using the generalized reduced gradient method.
(1)$$B_{i}=C_{i}\cdot K$$
(2)$$K=\frac{H_{20}\cdot H_{20}}{S_{2}\cdot S_{2}}+\frac{H_{20}\cdot H_{49}\cdot S_{42}}{S_{2}\cdot S_{49}}+\frac{H_{49}\cdot H_{49}}{H_{20}\cdot S_{42}\cdot S_{49}}$$

The next stage revealed the weight coefficients *C_i_*, at which the square of the deviation of *B_i_* from the value of *K* multiplied by *C_i_* is minimal. Finally, the model was tested, at which the constant error was corrected by subtracting the error, since the calculated values were always obtained by about 5% more (3).
(3)$$B_{i}=C_{i}\cdot \left(\frac{H_{20}\cdot H_{20}}{S_{2}\cdot S_{2}}+\frac{H_{20}\cdot H_{49}\cdot S_{42}}{S_{2}\cdot S_{49}}+\frac{H_{49}\cdot H_{49}}{H_{20}\cdot S_{42}\cdot S_{49}}\right)\cdot \left(1-0.054\right)$$

The model was developed based on model solutions of normal blood serum (SPINREACT normal), while model solutions of pathological serum (SPINREACT pathologic) were used for testing. The concentrations of all components in the test model solution corresponded to the concentration range of the initial model solutions.

## 3. Results

At the first stage, the FTIR spectra of serum and the proposed model solutions were compared (Figure 1). Figure 1A shows a typical FTIR spectrum of blood serum from a healthy volunteer. It was shown that the contributions of proteins to FTIR spectra are in the frequency ranges 3400–3030 cm^−1^, 1720–1480 cm^−1^, and 1301–1229 cm^−1^ (region I); lipids are in the ranges 3020–2819 cm^−1^, 1750–1725 cm^−1^, and 1480–1430 cm^−1^ (region II); carbohydrates and nucleic acids (DNA/RNA) are in the frequency range 1200–900 cm^−1^ (region III, Figure 1A). These can be briefly described as 3280 cm^−1^ (H-O-H stretching), 2957 cm^−1^ (asymmetric CH_3_ stretching), 2920 cm^−1^ (asymmetric CH_2_ stretching), 2872 cm^−1^ (symmetric CH_3_ stretching), 1650 cm^−1^ (amide I of proteins), 1536 cm^−1^ (amide II of proteins), 1453 cm^−1^ (CH_2_ scissoring), 1394 cm^−1^ (C=O stretch of COO^−^), 1242 cm^−1^ (asymmetric PO_2_ stretch), 1171 cm^−1^ (ester C–O symmetric stretch) and 1080 cm^−1^ (CO stretch). In region IV, vibrations of some functional groups of proteins and lipids were also found (Figure 1A).

Comparison of serum spectra in the 1800–1000 cm^−1^ region with a series of standards of individual substances (albumin, urea, glucose, etc.) at their normal serum concentration levels indicates a great similarity between the spectra of serum and model solutions and albumin, which indicates that the main contribution is from proteins (70% of the non-aqueous portion of the serum) for all spectra. Useful identification bands could be observed for urea at 1630 and 1460 cm^−1^ and for glucose between 960 and 1180 cm^−1^. Typical lipid bands can be identified in the wavenumber range 1700–1800 cm^−1^ and between 2600 and 3000 cm^−1^. Uric acid has a specific band at 1570 cm^−1^, and creatinine has two intense bands at 1720 and 1556 cm^−1^. However, typical bands of individual compounds were clearly observed in FTIR spectra only at concentration levels 10–100 times higher than those corresponding to normal values. Since the samples are complex mixtures of several compounds with many functional groups, the spectra are a complex set of FTIR absorption bands.

It should be noted that the maximum differences between the FTIR spectra of serum and model solutions are observed in regions III and IV and are due to the absence of a number of carbohydrates and nucleic acids in model mixtures (Figure 1B).

At the next stage, FTIR spectra of model solutions with different concentrations of constituent substances were obtained (Figure 2). It was shown that the intensities of absorption bands change ambiguously upon dilution of model solutions (Figure 2A). The FTIR spectra are strongly dominated by a large number of proteins contained in the serum, which are present in high concentrations in comparison with other low molecular mass components. In fact, the peak of amide I at 1650 cm^−1^ has the highest intensity in the entire spectrum. It is shown that the position of the absorption bands remains unchanged during the dilution process (Figure 2A). However, some of the absorption bands in the FTIR spectra of model solutions become more intense with a decrease in the protein content.

To construct the regression equation, four absorption bands were selected: H20 and H49—absorbance of the absorption bands at 1717 and 3903 cm^−1^; S2, S42, and S49 are the areas of the absorption bands at 616, 3750, and 3903 cm^−1^ (Figure 2). Figure 2B–E show the change in absorption bands with a change in the concentration of model solutions.

When the corresponding solutions are diluted, the intensities and areas of the selected absorption bands decrease (Figure 3).

The absorption bands selected in the construction of the model are not previously mentioned in the literature for semi-quantitative or quantitative analysis of blood serum. The absorption band at 616 cm^−1^ can be attributed to the ring deformation of phenyl [16,17]. The absorption band at 1717 cm^−1^ corresponds to the vibrations of the C=O bond amide I (arises from C=O stretching vibration), DNA, RNA, and purine base [18]. The absorption band at 3750 cm^−1^ can be attributed to the vibrations of the free OH-group, while no correspondences were found for the 3903 cm^−1^ band.

The coefficients (*C_i_*) for the calculation are given in Table 1.

Table 2 shows a complete list of the determined components of the model solution, the concentration range that was used in constructing the model, as well as the true model concentration of the test solution and that found during testing. According to the data presented, in all cases, the error in determining the concentration did not exceed 0.1% (Table 2).

## 4. Discussion

Methods for determining a number of serum parameters using FTIR spectroscopy have already been developed for blood [19,20,21]. Thus, blood glucose, an important parameter for the control of diabetes and other common diseases, has been determined in serum with acceptable accuracy [22,23,24]. It has been shown that spectroscopy in the mid-IR range can serve as an alternative basis for the clinical measurement of urea and glucose in blood serum [25]. The literature reports promising results on important serum parameters such as urea, total protein, albumin, triglycerides or total cholesterol [26,27,28,29]. Nevertheless, the possibility of simultaneous determination of several blood parameters has not been studied enough. For example, analysis of dry serum deposits using transmission spectroscopy revealed the possibility of quantitative determination of eight serum analytes [30] and simultaneous determination of malaria parasitemia, glucose and urea [31]. A reagent-free method for the simultaneous and direct detection of three analytes in human blood (glucose, triglycerides, and total cholesterol) based on FT-Raman spectroscopy has been proposed [32]. This study showed that the potential for quantitative determination of blood serum parameters is much wider, in particular, it is possible to simultaneously determine a larger number of analytes with greater accuracy.

In light of routine clinical laboratory use, relative prediction errors can be compared with standard deviations of reference concentrations, which primarily reflect physiological changes in the population [33]. Thus, the standard deviation of the reference values for total protein is 5.4% of the average concentration, while according to FTIR spectroscopy the protein concentration can be predicted with a relative prediction error of 4.7%. Similar conclusions can be drawn for HDL-cholesterol and uric acid, for which the relative prediction errors exceed the biological variability among donors of the studied population by 30% or less. In contrast, the variation in values for cholesterol, triglycerides, LDL-cholesterol, and urea is four times less than the biological variation in concentrations. It is for these parameters that the mid-infrared range can be a valuable quantification tool. On average, the accuracy of the determination ranges from 4% for total protein to 16% for LDL-cholesterol [33]. Another study provides comparable values for the accuracy of determining a number of blood analytes: total protein—2.2%, albumin—4.0%, glucose—18.5%, urea—19%, total cholesterol—15.7% [3]. The least accurate is the determination of the triglyceride content by FTIR spectroscopy (error 43.1%) [3]. The low values of the error in the determination of all analytes obtained by us are due to the fact that solutions of normal serum with a balanced average ratio of analytes were taken for modeling, while in real serum samples deviations in the content of individual components are possible even within the normal range, including extreme values against the background of various diseases. When switching to real blood serum samples, there will undoubtedly be a loss of accuracy, however, low error values on the model system provide its reserve.

Overall, infrared spectral datasets are rich in information, highlighting underlying biological and structural differences. Combined with powerful multivariate analysis approaches, they can distinguish between disease classes by extracting relevant information. Spectral data analysis has used a variety of data mining approaches such as principal component analysis (PCA), random forest (RF), and support vector machine (SVM), all of which demonstrate the ability to distinguish patients from non-diseased biofluid samples [34]. Least squares regression analysis (PLSR) is currently one of the most commonly used methods for quantitative modeling due to its ability to detect systematic variation in influencing factors and generate quantitative predictive models [35]. This allows unknowns to be predicted using hidden variables extracted from the regression model [36,37]. A robust regression model is described for establishing multivariate calibration based on a nonlinear iterative partial least squares (NIPALS) algorithm with orthogonal signal correction (OSC) and sample set splitting based on joint distance x-y (SPXY) [32].

Thus, FTIR spectroscopy is able to detect minor differences in biofluid samples with minimal sample preparation, and numerous studies supporting the principle have highlighted the potential clinical use of this method. However, broad uptake of FTIR spectroscopy has not happened due to a variety of factors, including a lack of acceptance from the clinical environment.

In this study, we have demonstrated for the first time the possibility of simultaneous determination of 38 parameters in blood serum. The concentration range of the components corresponds to their real concentration in the blood serum. This study is preliminary and has a number of limitations. Modeling was carried out on systems in which the ratio between them did not change when the concentration of the components changed. In real clinical samples, this consistency will not be observed. At the next stage of the study, we plan to analyze model systems with different ratios of constituent substances, as well as with an extreme content of individual components, which is observed in various pathologies. We assume that the results obtained will require changes to the model. Only after that will the proposed model be tested on real clinical blood serum samples. At the same time, we predict a significant decrease in the accuracy of determining the concentrations of individual components of serum. It should be noted that even in the case of successful approbation and validation of the method, its clinical use is possible mainly for express analysis and in any case will require additional confirmation by standard laboratory methods.

## 5. Conclusions

Modeling of normal blood serum was carried out. It was shown that using the absorbance of the absorption bands at 1717 and 3903 cm^−1^ and the areas of the absorption bands at 616, 3750, and 3903 cm^−1^, it is possible to simultaneously determine the concentrations of 38 components with an error of less than 0.1%. The results obtained confirm the potential clinical use of FTIR spectroscopy as a reagent-free express method for the analysis of blood serum. However, its practical implementation requires additional research, in particular, analysis of real blood serum samples and validation of the method.

## Figures and Tables

**Figure 1 diagnostics-11-02391-f001:**
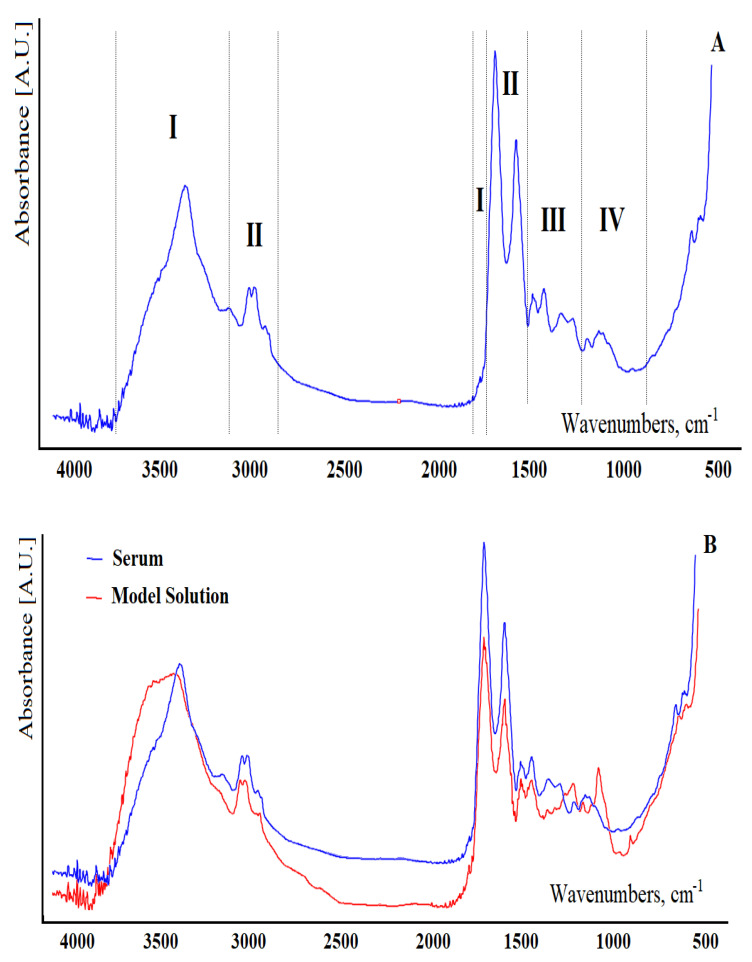
FTIR spectra of the serum (**A**) and model solution (**B**). Region I—absorption of proteins (3400–3030 cm^−1^, 1720–1480 cm^−1^, and 1301–1229 cm^−1^); region II—lipid absorption (3020–2819 cm^−1^, 1750–1725 cm^−1^ and 1480–1430 cm^−1^); region III—absorption of carbohydrates and nucleic acids (DNA/RNA) in the range 1200–900 cm^−1^; region IV—vibrations of some functional groups of proteins and lipids.

**Figure 2 diagnostics-11-02391-f002:**
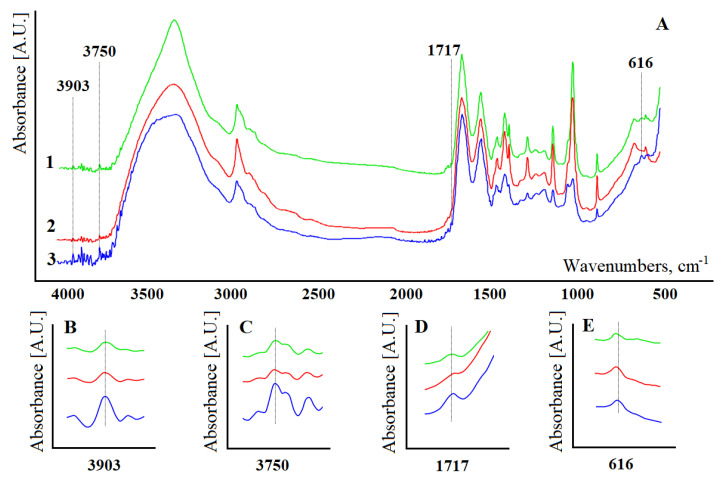
FTIR spectra of the model solutions when the stock solution is diluted 5 times (curve 1), 3 times (curve 2), and 2 times (curve 3), respectively (**A**). Enlarged portions of FTIR spectra for 3903 cm^−1^ (**B**), 3750 cm^−1^ (**C**), 1717 cm^−1^ (**D**) and 616 cm^−1^ (**E**).

**Figure 3 diagnostics-11-02391-f003:**
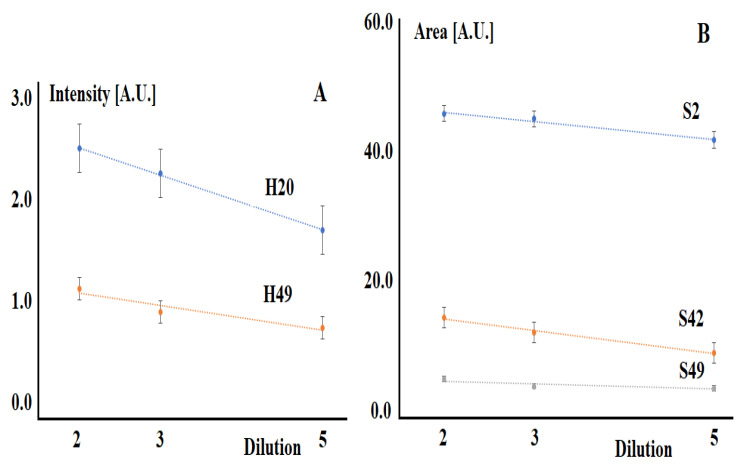
Changes in the absorbance (**A**) and areas (**B**) of characteristic absorption bands upon dilution of model solutions. The correlation coefficients between the intensities and areas of absorption bands and dilution are equal for H20—0.9992, H49—0.9165, S2—0.9762, S42—0.9887, S49—0.7719, *p* < 0.05.

**Table 1 diagnostics-11-02391-t001:** Coefficients (*C_i_*) for calculating the concentrations of components *B_i_* (*i* = 1...38).

Component	B1	B2	B3	B4	B5	B6	B7	B8	B9	B10
*C_i_*	1019	136.7	61.42	458.7	1507	21.46	5.474	32.66	7.744	2.815
Component	B11	B12	B13	B14	B15	B16	B17	B18	B19	B20
*C_i_*	2.768	4.447	3.685	5.474	167.9	51.32	65.93	106.8	37.16	3.81
Component	B21	B22	B23	B24	B25	B26	B27	B28	B29	B30
*C_i_*	2.084	227	3.623	531.8	78.53	53.8	317.2	768.2	8554	88.48
Component	B31	B32	B33	B34	B35	B36	B37	B38		
*C_i_*	329.7	309.4	247.2	220.8	131.4	1157	65.47	489.8		

**Table 2 diagnostics-11-02391-t002:** List of components and concentration range of model solutions.

№	Component	Units	Concentration Range *	Test (True)	Test (Found)	∆, %
B1	Uric acid	μmol/L	58.40–292.00	131.00	130.96	0.03
B2	Total bilirubin	μmol/L	5.78–28.90	17.58	17.57	0.06
B3	Direct bilirubin	μmol/L	3.06–15.30	7.90	7.89	0.08
B4	Creatinine	μmol/L	18.02–90.10	59.00	58.95	0.08
B5	Fructosamine	μmol/L	147.40–737.00	193.80	193.68	0.06
B6	Glucose	mmol/L	1.04–5.22	2.76	2.76	0.07
B7	Lactate	mmol/L	0.30–1.52	0.70	0.70	0.07
B8	Urea	mmol/L	1.39–6.97	4.20	4.20	0.06
B9	Cholesterol	mmol/L	0.48–2.39	1.00	1.00	0.07
B10	HDL-Cholesterol	mmol/L	0.14–0.72	0.36	0.36	0.06
B11	LDL-Cholesterol	mmol/L	0.27–1.34	0.36	0.36	0.07
B12	Phospholipids	mmol/L	0.32–1.60	0.57	0.57	0.08
B13	Triglycerides	mmol/L	0.25–1.24	0.47	0.47	0.08
B14	Calcium	mmol/L	0.44–2.22	0.70	0.70	0.07
B15	Chlorides	mmol/L	16.20–81.00	21.60	21.58	0.10
B16	Copper	μmol/L	3.64–18.20	6.60	6.60	0.06
B17	Iron	μmol/L	3.68–18.40	8.48	8.47	0.08
B18	Total Iron binding capacity	μmol/L	13.74–68.70	13.74	13.73	0.10
B19	Potassium	mmol/L	0.70–3.51	4.78	4.78	0.09
B20	Lithium	mmol/L	0.20–1.00	0.49	0.49	0.07
B21	Magnesium	mmol/L	0.17–0.87	0.27	0.27	0.06
B22	Sodium	mmol/L	25.20–126.00	29.20	29.18	0.09
B23	Phosphorus	mmol/L	0.27–1.36	0.47	0.47	0.08
B24	Zinc	μg/mL	77.40–387.00	68.40	68.35	0.08
B25	Total Protein	g/L	13.24–66.20	10.10	10.09	0.07
B26	Albumin	g/L	9.78–48.90	6.92	6.91	0.08
B27	α-amylase	U/L	16.40–82.00	40.80	40.77	0.08
B28	Creatinekinase	U/L	29.00–145.00	98.80	98.73	0.07
B29	Cholinesterase	U/L	1107.80–5539.00	1100.20	1099.39	0.07
B30	Acid Phosphatase	U/L	6.72–33.60	11.38	11.37	0.07
B31	Alkaline Phosphatase	U/L	15.20–76.00	42.40	42.37	0.06
B32	Gamma glutamyltransferase	U/L	8.40–42.00	39.80	39.77	0.09
B33	Asparagine aminotransferase	U/L	10.60–53.00	31.80	31.77	0.09
B34	Alanine aminotransferase	U/L	10.40–52.00	28.40	28.38	0.08
B35	Lipase	U/L	7.86–39.30	16.90	16.89	0.07
B36	Lactate dehydrogenase	U/L	78.00–390.00	148.80	148.70	0.07
B37	Glutamate dehydrogenase	U/L	4.98–24.90	8.42	8.41	0.07
B38	α-hydroxybutyrate dehydrogenase	U/L	31.00–155.00	63.00	62.95	0.08

Note. *—The concentration range of the components corresponds to their real concentration in the blood serum.

## Data Availability

Not applicable.

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
