# Peer review of "Application of FTIR Spectroscopy for Quantitative Analysis of Blood Serum: A Preliminary Study"

_diagnostics, 2021, doi:10.3390/diagnostics11122391_

Round 1
Reviewer 1 Report
In the manuscript "Application of FTIR spectroscopy for quantitative analysis of blood serum: A preliminary study" the authors present a mathematical model to predict the amount of blood content from FTIR measuremets. The paper fits very well in the Diagnostics Journal and can be published after some improvements.
The authors need to compare the accuracy of the presented results with other similar studies using FTIR.
The typical blood serum spectrum shown in figure 1A is from a healthy volunteer? Please, explain more the differences between the serum and the model. If the serum had some alterations (for example, more or less HDL) the variation will be significant? The concentrations range used in table 2 are the ones expected in real blood samples?
Author Response
The authors express their gratitude to the reviewers for their attentive attitude and careful consideration of the manuscript.
1. The authors need to compare the accuracy of the presented results with other similar studies using FTIR.
We've added the following information to the Discussion section:
«In light of routine clinical laboratory use, relative prediction errors can be compared with standard deviations of reference concentrations, which primarily reflect physiological changes in the population [33]. Thus, the standard deviation of the reference values ​​for total protein is 5.4% of the average concentration, while according to FTIR spectroscopy the protein concentration can be predicted with a relative prediction error of 4.7%. Similar conclusions can be drawn for HDL-Cholesterol and uric acid, for which the relative prediction errors exceed the biological variability among donors of the studied population by 30% or less. In contrast, the variation in values ​​for cholesterol, triglycerides, LDL-Cholesterol, and urea is four times less than the biological variation in concentrations. It is for these parameters that the mid-infrared range can be a valuable quantification tool. On average, the accuracy of the determination ranges from 4% for total protein to 16% for LDL-Cholesterol [33]. Another study provides comparable values ​​for the accuracy of determining a number of blood analytes: total protein - 2.2%, albumin - 4.0%, glucose - 18.5%, urea - 19%, total cholesterol - 15.7% [3]. The least accurate is the determination of the triglyceride content by FTIR spectroscopy (error 43.1%) [3]. The low values ​​of the error in the determination of all analytes obtained by us are due to the fact that solutions of normal serum with a balanced average ratio of analytes were taken for modeling, while in real serum samples deviations in the content of individual components are possible even within the normal range, including extreme values ​​against the background of various diseases. When switching to real blood serum samples, there will undoubtedly be a loss of accuracy, however, low error values ​​on the model system provide its reserve.»
2. The typical blood serum spectrum shown in figure 1A is from a healthy volunteer? Please, explain more the differences between the serum and the model. If the serum had some alterations (for example, more or less HDL) the variation will be significant? The concentrations range used in table 2 are the ones expected in real blood samples?
Figure 1A shows a typical FTIR spectrum of blood serum from a healthy volunteer. The concentration range of the components in Table 2 corresponds to their real concentration in the blood serum. Nevertheless, in the continuation of research, we plan to consider options with an extreme content of individual components characteristic of various pathologies. We've added this limitation to the discussion section.
Reviewer 2 Report
The paper entitled "Application of FTIR spectroscopy for quantitative analysis of 2 blood serum: A preliminary study" is an incomplete attempt in strict scientific language. The study in not supported by plausible instrumental evidences and no comparison has been made in this regard.
- In the abstract specifications of serum and FTIR machine is unlikely, revise it.
- Table 2, Column 2 is not needed the unit may be mentioned in caption of the table and rare unit shall be specified in foot note or in the caption together with the unit of majority of samples.
- There is no balance between sections (in terms of volume), particularly results and discussion. Discussion of the study has been summarized in one page which is too short and contains only citations from literature. Its like introduction of the work being carried out.
- At the end of Discussion section the authors declare that the study is incomplete, it has not been applied for analysis of real biological samples. It indicates that the study has least scientific impact and acceptance.
- It is suggested that the authors must improve the discussion, compare their results of the same samples measured with already in-field instruments. After critical comparison, the study will become easily understandable. As they mentioned in their article that FT-IR analysis is not a practical technique in the field and still there are endless questions to convince the scientific community regarding results particularly exact concentration of constituents of a complex mixture like blood. This technique suffers from many limitations in measuring very simple chemical mixtures.
This reviewer is not really convinced with the study and will favor its acceptance after collecting results with the help of other techniques and their critical comparison.
Reviewer 3 Report
The topic of this manuscript is certainly interesting for the scientists who work in the medical diagnostic field since the serum and whole blood are the most important and the most frequently used diagnostic materials. Therefore, searching for new methods to monitor or quantify the minor content of biologically active compounds in blood, urine, saliva and other diagnostic materials is very desirable. Furthermore, I agree with the idea that spectroscopic methods, especially FT-IR, have an enormous potential in the medical diagnostic field. Thus, the manuscript deserves to be published in Diagnostics, but after some minor corrections.
According to the authors statements in the lines 197-199, the clinical use of the developed procedure must be preceded by analysis of real serum samples and validation of this method. It should be clearly indicated in the Abstract and Conclusions.
line 120 – “low molecular mass compounds” instead of “low molecular weight components”
2.3. Regression model development – this section is too generally described. Please, give more details, for example – what is “the algorithm” mentioned in line 145?
Figure 3 – What were the correlation coefficients between intensity and dilution, and between area and dilution? The straight lines seems to be very flat.
line 149 – “purine base [18].” instead of “purine base18.”
Table 2 – component number B38 is missing in this table
Reviewer 4 Report
The manuscript titled “Application of FTIR spectroscopy for quantitative analysis of blood serum: A preliminary study” concerns the study of analyzing the possibility of simultaneous determination of the concentration of components from the characteristics of FTIR spectra using the example of a model blood serum. To prepare model solutions, a set of freeze-dried control sera based on bovine blood serum was used, certified for approximately 38 parameters. As the result authors confirm possibility to simultaneously determine the concentrations of 38 components with an error of less than 0.1%.
Described method seems to be useful in far future, rather cost effective. The manuscript touches important analytical topic. The paper is rather well-written, however it fails in some points. It fails in demonstrating the advantages of the investigated setup with respect to similar schemes. Authors should emphasize the benefits of their development according to these known from literature. The presented analytical strategy is difficult to interpret the result, the authors should clearly discuss its limitation, show what their work should focus on. They should show where to look for differences in working with the model and the real sample. Due to the fact, manuscript may be ready for publication after major revision.
Round 2
Reviewer 2 Report
The author have addressed almost all the issues and now their paper is very clear regarding the applicability of the technique for clinical use. FT-IR for determination/quantification of a large number and variety of components in blood serum will still remain a dream. Please check the sentence, "FTIR spectroscopy (error 43.1%) [3]. The low values of.......". 43.1% error is not low. In the same paragraph the sentence at the end may be clarified (.......model system provide its reserve.).
This reviewer is satisfied from the changes they made and recommends the paper for publication in diagnostics.
Reviewer 4 Report
The manuscript titled “Application of FTIR spectroscopy for quantitative analysis of blood serum: A preliminary study” concerns the study of analyzing the possibility of simultaneous determination of the concentration of components from the characteristics of FTIR spectra using the example of a model blood serum. To prepare model solutions, a set of freeze-dried control sera based on bovine blood serum was used, certified for approximately 38 parameters. As the result authors confirm possibility to simultaneously determine the concentrations of 38 components with an error of less than 0.1%.
Described method seems to be useful in far future, rather cost effective. The manuscript touches important analytical topic. The paper is rather well-written, and the revised form may be ready for publication.